# Experimental Study on the Possibilities of FDM Direct Colour Printing and Its Implications on Mechanical Properties and Surface Quality of the Resulting Parts

**DOI:** 10.3390/polym14235173

**Published:** 2022-11-28

**Authors:** Ioan Tamașag, Cornel Suciu, Irina Beșliu-Băncescu, Constantin Dulucheanu, Delia-Aurora Cerlincă

**Affiliations:** 1Faculty of Mechanical Engineering, Automotive and Robotics, Stefan cel Mare University, 720229 Suceava, Romania; 2Faculty of Electrical Engineering and Computer Science, Stefan cel Mare University, 720229 Suceava, Romania

**Keywords:** additive manufacturing, colour 3D printing, surface quality, mechanical proprieties

## Abstract

The present paper aims to contribute to the methodology of 3D printing in-process colouring and study its implications and impact on the tensile strength and surface quality of the obtained parts. The proposed study was based on a Taguchi L27 DOE plan using standardised EN ISO 527-2 type 1B-shaped specimens, in which four factors on three levels were considered. The obtained results highlight the possibility of using the presented in-process colouring method. Different materials (PLA, PLA+, and PETG) with varying infill densities (15%, 30%, and 50%), colour distribution (33%, 66%, and 99%), and colour pigments (blue, green, and red) were studied and the results highlighted that the most influential parameter on the tensile strength of the parts was infill density, followed by the tested material, colour pigment, and colouring percentage; regarding surface roughness, the most influential parameter was infill density, followed by colouring percentage, colour pigment, and material. Moreover, the values resulting from the Taguchi DOE were compared to uncoloured parts, from which it could be concluded that the colouring of the parts had direct implications (negative for tensile strength and positive for surface roughness).

## 1. Introduction

With the intensive use of additive manufacturing in increasingly different fields [1], whatever its nature (MEX—material extrusion, VPP—vat photopolymerization [2], etc.), the attention of researchers has been focused on improving the process through various methods, such as post-processing of the parts obtained to increase mechanical performance, improving surface quality, or offer new capabilities to the process, such as the possibility of true colour reproduction. Traditionally, the colouring of 3D-printed parts, especially in the case of MEX manufacturing, was carried out by mechanical post-processing. The necessary steps for conventional post-process colouring are shown schematically in Figure 1. 

On the topic of part colour and its study, several studies have been conducted [1,3,4,5,6,7] where the main focus of the study was the possibility of colouring 3D-printed parts regardless of the type of additive manufacturing.

However, in the field of additive manufacturing, research has been directed towards the possibility of reducing the time and resources consumed by post-processing operations, which has involved attempts to increase the mechanical and physical parameters resulting directly from the manufacturing process in the in-process form. In the case of in-process colouring during additive manufacturing, a number of studies have emerged [8,9] that classify, in detail, the methodology of obtaining colour.

In the case of FDM printing, these methods include using multiple print heads and colouring the previously deposited layer using the inkjet method (Figure 2a). More recently in the 3D printing community, colouring the filament before extruding has been proposed (Figure 2b).

Numerous studies have been carried out regarding both the mechanical properties and surface quality of 3D-printed parts in the MEX process. The main issues studied were the manufacturing input factors (temperature, infill, printing direction, etc.) [10,11,12,13], post-processing operations (heat or chemical treatments) [14,15,16,17,18,19,20], and material structure or composition [21,22,23].

The literature presents a number of studies on filament colour and its implication on specific parameters. For instance, in [24], the author studied the influence of polylactic acid (PLA) colour on thermal behaviour and surface quality. Thus, a strong correlation between the pigment used and the properties of the 3D-printed parts was reported. The results showed that, using filaments of different colours (natural, green, black), the surface quality and regularity of layers highly differed depending on the colour. Additionally, they concluded that each colour has an influence on thermal behaviour, and suggested using a different printing temperature depending on the filament colour. The best results, both in terms of surface quality and temperature behaviour, were obtained for the PLA natural material. 

Another study [25] addressed the influence of colour on the tensile strength of PLA parts by using different colours of filament made by the same manufacturer. The study showed that the colour of PLA 3D-printed parts varies parameters significantly, with an 18% difference in terms of elastic modulus, 36% in terms of yield strength, and 31% in terms of ultimate tensile strength. 

Filament colour and extrusion temperature implications were also studied in [26], where the authors concluded that the pigment used to colour the filament has a significant influence on the crystallisation effect of the PLA material. It was thus considered that there is a critical temperature for optimising crystallisation that depends on the colour of the filament. 

Regarding the implications of in-process colouring on the process parameters of interest, the literature provides limited information. Therefore, the present study is novel and represents a research opportunity. The main novelty of this paper is represented by the use of a new device of original design and construction that allows for the implementation of a new in-process colouring method that uses alcohol-based ink. The impact of in-process colouring on mechanical properties and surface quality was investigated for three of the most commonly used materials in additive manufacturing.

Using this approach, the aim of the paper is to analyse the influence of in-process filament colouring (Figure 2a) on tensile strength, impact strength, and surface quality from a geometrical point of view (roughness). Given the current state of the art and the fact that the main materials used in MEX fabrication are PLA (polylactic acid), PLA+ (a superior version of PLA obtained by using various additives), ABS (Acrylonitrile Butadiene Styrene) [27], and PETG (polyethylene terephthalate glycol) [28], a Taguchi L27 experimental design was used for this study, in which three materials (PLA, PLA+, and PETG), three different coloured inks (red, green, and blue), three colouring percentages (~33%, ~66%, and ~99%), and three infill ratios (15%, 30%, and 50%) were varied. The materials used in the present study are described in Section 2.1. The tests were conducted in accordance with the ISO 527 standard: Plastics—Determination of tensile properties, part 1 and 2 [29,30].

The experimental results presented below show that in the case of tensile force, in-process filament colouring has a negative impact, while in-process filament colouring has a positive impact in the case of surface roughness, as roughness values improve substantially.

## 2. Experimental Setup

To achieve the proposed objectives, a DOE Taguchi experimental design (shown in Table 1) was used, and the experimental program illustrated by Figure 3 was followed.

Figure 3 graphically illustrates the steps taken for the present experimental study, which are further described in the following sections. Two types of specimens were printed, one for tensile tests and one for impact strength testing. Tensile tests were conducted using specialised equipment, and Charpy tests were performed using a Galdabini Impact 300 pendulum hammer. Surface micro-topographies of test specimens were mapped using a Mahr MarSurf CWM 100 confocal microscope and interferometer, and surface quality parameters were determined. Optical images of the surfaces were taken with the aid of an “Optech” model IM/IMT microscope and Optika Vision Pro image analysis software. All the abovementioned experimental steps and the equipment used are further described in Section 2.2.

The results in Table 1 were compared with the values obtained from the tests on uncoloured specimens shown in Table 2.

### 2.1. Materials

This paper analyses the MEX in-process colouring capabilities of the three most-used 3D printing materials: PLA, PLA+, and PETG. 

The investigated materials were produced by the same manufacturer (eSUN—Shenzhen Esun Industrial Co., Ltd., based in Shenzhen, China), and their properties according to technical data sheets [31,32,33] are summarised in Table 3.

The chemical compositions of the used materials, according to the manufacturer, are as follows: PLA = 99.8% polylactic acid resin [34]; PLA+ = 92–96% polylactic acid resin, 2–4% calcium carbonate powder, and 2–5% other additives [35]; PETG = > 95% polyester and <10% other additives [36].

Besides being an environmentally friendly material, PLA presents high strength, low thermal expansion, and high heat resistance, with one of the highest melting points among biodegradable polymers. All these characteristics recommend this material for a wide range of applications. PLA+ refers to the superior versions of polylactic acid that were developed in order to overcome some of the drawbacks of PLA, such as brittleness and moisture retention. Usually, PLA+ is obtained by polycarbonate fortification of PLA. PETG stands for polyethylene terephthalate glycol. This material is used in the food industry and for some medical applications. The main proprieties that recommend PETG are superior shock resistance, chemical stability, hardness, and ductility. The ink used for the printing process was a highly saturated, fast-drying, highly transparent alcohol-based ink produced by Jacquard, USA, in accordance with ASTM D-4236. 

### 2.2. Sample Preparation and Equipment

The specimens used were in the shapes shown in Figure 4, where the specified dimensions are in mm (the metric system was used). For the tensile strength study, specimens with the standardised ISO 527 type 1B shape (Figure 4a) were used. For the impact strength study, specimens with the standardised shape shown in Figure 4b were used.

The specimens were printed using a budget 3D printer model (CR6-Se) produced by Creality (Figure 5), which was equipped with a specially designed filament colouring device and a SUNLU filament dryer. Before use, the filament was dried for 5 h at 40 °C. 

The colouring of parts was performed using the principle presented in Figure 6a using a device of our own design and construction, which is presented in Figure 6b. Using this device, coloured inks are applied to the filament surface before it enters the hot end. Inside the nozzle, due to pressure and high temperature, the ink merges with the uncoloured filament, resulting in coloured parts. 

When designing the device, particular attention was paid to its dimensions. We tried to compact the components so as not to influence the dimensional capacity of the 3D printer. The device, made entirely of PLA material insoluble in alcohol or water, was placed on the printer mentioned above, with the proviso that it could be used universally on other types of 3D printer. The housings allow the tanks to be positioned in two ways (Figure 6c): closed, when contact between the filament and the porous material occurs, and open, when there is no contact between the two.

The whole device was placed on the 3D printer’s extrusion head and stiffened by a support. The PTFE (Teflon) tube of the 3D printer (6), through which the filament (7) passes, is assembled with the base of the device (1) by means of standardised dies (5). The construction of the device base allows three housings (2) to be mounted through a standardised ‘dovetail’ channel, all located equidistantly at an angle of 120°. Ink reservoirs (3), consisting of a porous material impregnated with ink (8), are mounted in the housings (2), with an outlet at the top only, and are sealed with threaded covers (4). The proposed new device is mounted on the 3D printer using a socket (5) and tube (6). The filament (7) passes through the base of the device (1) and is coloured when the ink cap (8) inside the reservoir (3) is in the ‘closed’ position. Depending on the intensity of the desired colour in contact with the filament, one, two, or three reservoirs may be in contact with the filament at the same time.

The device allows for the simultaneous use of 3 colours, making it possible to study the percentage of colouring, as shown in Figure 7. 

Since previous studies [37] have shown the influence of the position of the nozzle at the beginning of each layer (Z seam), all parts were made so that the beginning of the layer was not randomly located and thus did not influence the test area of the parts (see Figure 8), an option modified in the Cura 5.0 slicer. The process parameters remained constant for PLA parts and the extrusion temperature was changed for PETG parts (Table 4). Additionally, the part was printed flat, with the printing direction for both tensile testing specimens and Charpy specimens running along the X axis.

The tensile strength of the specimens (Figure 9b) was measured using an experimental setup (Figure 9a) previously built within the Faculty of Mechanical Engineering, Automotive, and Robotics at the “Stefan cel Mare” University of Suceava, Romania. The setup is composed of a clamping device (1) mounted on the tensile testing device (2), which in turn is operated by a controller (4). The specific elongation and traction force values were recorded using a sensor (3) and an amplifier (5). The experimental data were processed and analysed on a PC (6). 

Impact energy tests were performed using an industrial Galbadini Impact 300 pendulum hammer (Figure 10).

Sq surface roughness values were obtained using the Mahr MarSurf CWM 100 confocal microscope and interferometer (Figure 11a), and surface topography (Figure 11b) was analysed using the related MountainsLab 8.1 software. Sq is a surface texture parameter (ISO 25178) equivalent to the standard deviation of heights and is calculated as the root mean square value of ordinate values within the definition area.

Measurements were taken in the central region of the specimens in 2 × 1.5 mm^2^ areas. Surface roughness was a consideration, thus both directions were taken into account. The parameters were determined in accordance with the ISO25178 standard with the default MountainsLab 8.1 software settings, which use a predefined 0.8 mm robust Gaussian L-filter.

Surface analysis was also performed using an “Optech” microscope (Figure 12), model IM/IMT (manufacturer: Exacta + Optech GmbH, München, Germany), and surface analysis was performed using an image acquisition (optical microscopic analysis) software called Optika Vision Pro. 

## 3. Results and Discussion

Through application of the experimental design shown in Figure 3 and with the capabilities of the equipment used, the possibility of in-process filament colouring was demonstrated; the values obtained from the experimental tests are listed in Table 1. Based on these results, mean effect variance plots were obtained and ANOVA analysis was applied to determine the degree of influence that the studied factors had on the output parameters. 

### 3.1. Tensile Tests

Regarding the experimental results obtained from the tensile tests of the parts, a good correlation with the literature is observed, especially in the case of the percentage of infill and the material used. Figure 13 shows the variation of the main effects of the studied parameters for the tensile strength tests. 

Regarding the studied material, it can be seen that the best values were recorded with the PLA+ material, followed by PLA and finally PETG. As in the studies carried out by the authors of articles [24,25], it was observed that pigment had an influence on the mechanical properties of the printed parts. The red pigment had a negative impact on tensile strength, while the blue pigment was shown to give better values. Obviously, by increasing the percentage of infill, the tensile strength of the parts also increased. In terms of the percentage of colouring, the values obtained are almost linear in shape, with the lowest results being recorded when the filament was 99% coloured. This can be explained by the fact that the amount of ink in the final filament composition increased. The percentage values of variation are listed in Table 5, where 1, 2, and 3 have the meaning from Table 1 (1—PLA, 2—PLA+, 3—PETG; 1—blue, 2—red, 3—green; 1—33%, 2—66%, 3—99%; and 1—15%, 2—30%, and 3—50%).

Comparing results obtained from the application of the Taguchi experimental design with results obtained for the parts without colouring (Table 6), it can be observed that the maximum variation of the values is decreasing. 

The ANOVA analysis allowed us to draw a Pareto chart (Figure 14), which shows the level of significance of each factor used in the experimental design presented in Section 2. 

Taking into account only the maximum variation, it can be considered that the application of in-process filament colouring negatively influences the tensile strength of the parts. However, in Figure 15, variation plots were drawn for all the values obtained and it was observed that the parts without colour occupy a median position. Therefore, in-process colouring does not affect tensile strength, as can also be seen from the ANOVA analysis.

Among the factors studied in the Taguchi experimental design and the data presented in Figure 13 and Figure 14, it can be seen that the factors with a statistically significant level (*p*-value < 0.05) are the percentage of infill and the material used. It can be considered that in relation to tensile force the percentage of infill has an influence of 54.98%, the material used an influence of 29.71%, pigment an influence of 12.85%, and the percentage of staining an influence of 2.46%. 

### 3.2. Roughness Tests

The results for geometric surface quality (surface roughness) show important correlations with tensile force. The values shown in Figure 16 highlight that PLA+ material, in addition to increased tensile strength, also provides better roughness, confirming other studies in the literature [38]. Regarding the pigment used, it is observed that the best roughness was obtained in the case of the red pigment at a colouring percentage of 99%. This may be due to the fact that, by mixing the filament with the used ink, the material changes its properties, softens, and the adhesion between the layers is better, resulting in a lower roughness value. At the same time, due to the softening effect of the material, the mechanical properties decrease. Contrary to the results presented in Section 3.1, the surface quality grows with increasing infill percentage. This phenomenon can be attributed to the enhanced stiffness of the parts during printing and a decrease in the convective heat transfer coefficient with increasing infill. This second phenomenon is due to an increase in cooling time, as more material is present in the same space with increased infill, thus allowing for the material to better fill the gaps between adjacent layers and leading to a possible improvement in surface quality. 

The percentage values of variation are listed in Table 7, where the roughness value is taken into account. In this respect, using PLA+ material resulted in an improvement in the values obtained of 2.62% compared to PLA material, and using PETG material resulted in a 2.88% degradation in surface quality compared to PLA material. In the same way, both the pigment and the percentage of colour distribution on the filament and the infill have similar variations to those presented in the previous subchapter. Thus, the green and red pigment recorded better values for roughness and colour distribution had a positive influence, i.e., by increasing colour distribution, surface quality was improved by up to 6.15%. By increasing the percentage of infill, an improvement in surface quality of up to 10.56% was achieved.

Contrary to the previous case, when comparing results obtained after the application of the Taguchi experimental plan with results obtained for parts without colouring (Table 8), it can be observed that, in most cases, the maximum variation of the values is decreasing. In this sense, it can be considered that application of in-process filament colouring positively influences surface roughness in the case of PLA and PLA+ material. 

However, the values obtained for PETG show an increase in the roughness values. This may be due to the morphological structure of the material. Compared to PLA, a material with a semi-crystalline structure, PETG has an amorphous structure, which leads to further chemical reactions between the material and the plaster used. Additionally, the extrusion temperatures may have an influence; in the case of PLA it was 220 °C and in the case of PETG it was 240 °C.

This tendency can also be observed in Figure 17. In Figure 16 the uncoloured parts made of PLA and PLA+ presented bigger values for surface roughness Sq, while they were on the opposite side in the case of PETG, as the uncoloured parts presented good overall values. There is also a good correlation with the Taguchi values presented in Figure 16, where it can be seen that, overall, the red pigment had presented the smallest values for surface roughness, implying that the red pigment has stronger chemical reactions with the filaments.

The Pareto chart (Figure 18) shows the level of significance of each important factor used in the experimental design presented in Section 2. In the case of the ANOVA analysis on surface roughness, the results show that none of the factors studied is statistically significant, with all *p*-values above 0.05. However, the degree of influence of each factor can be highlighted as follows: infill, 43.81%; colouring percentage, 24.14%; pigment, 19.84%; and material, 12.21%.

Analysing the values, it can be concluded that when using a material with a semi-crystalline structure, colouring has a positive impact on surface roughness, while when using material with an amorphous structure, colouring has a negative impact on roughness of up to about 40% for both cases. 

### 3.3. Optical Surface Analysis

Microscopic analysis of the specimens was able to confirm the possibility of colouring the filament using the method presented (Figure 19). The microscopic analysis showed that the intensity of the colours depends on both the colouring percentage and the material used. In the case of PLA and PLA+ materials, the colours are more vibrant, while in PETG material the colours are blander. At the same time, the green pigment used with PETG had visual changes, i.e., the resulting colour after extrusion had different shades of green and total colour intensity was bluer. 

Another important aspect is shown in Figure 20, where the images represent the microscopic appearance of the lateral surface of the 3D-printed probes on the layer adhesion area. 

Figure 20a,b correspond to the image observed with the Optika metallographic microscope, and Figure 20c,d correspond to the images of rough test surfaces obtained with the white light interferometer. Due to the small amounts of trapped air that build up during the 3D printing process, most of the PETG samples generated during the experimental tests exposed more air pores along the printing layers than PLA and PLA+, as can be seen in the sample presented in Figure 20b,d. This phenomenon explains the low tensile strength of the coloured printed parts. 

### 3.4. Charpy Tests

The impact test results ranged from 0.3 to 0.4 J for coloured parts and 0.4 J for uncoloured parts. The experimental results cannot be analysed due to the small variation between the values, and further determinations are required which will be addressed in a future study. However, after the tests, it could be observed that, in most cases, the coloured parts showed brittle fractures (Figure 21).

## 4. Conclusions

3D printing is still a technology under development, with new materials and new methods still being tested. There is a strong tendency in many industrial fields to replace metal parts with plastics, polymers, and composites. 3D printing is one of the main technologies for manufacturing polymers and composites. When considering this replacement, it is important to analyse the working conditions and mechanical loads and to have information about the capability of 3D-printed parts to fulfil those requirements. The present paper offers information regarding the quality and mechanical strength of 3D parts generated by in-process colouring.

Considerable effort is being devoted to the development of new materials and technologies to explore the possibility of eliminating post-process operations through improvements to 3D-printed parts and their use as replacements for in-process operations. One of these in-process operations is colouring of the parts. In this regard, the present paper investigates a new possible in-process method of colour application before filament melting. 

Using a device of our own design, it was possible to study the influence of three pigments, three materials, and three types of colouring and infill density on the tensile strength and surface quality of 3D-printed parts. The conducted study confirms the possibility of replicating colours on 3D-printed parts, regardless of the material, using alcohol-based ink. It was also observed that the colouring of the filament has an influence on mechanical properties and surface roughness. 

As far as tensile strength is concerned, the performed tests highlighted that, of the varied factors, infill density had the highest significance (54.98% influence), followed by material nature (29.71% influence), colour pigment (12.85%), and colouring percentage (2.46%). Even though the ANOVA analysis suggested that, from the varied factors, colouring of the material did not have statistical significance, when values were compared to the ones obtained for uncoloured parts, a deviation of 34.5% could be observed in the case of PETG, a deviation of 15.24% in the case of PLA, and a deviation of 11.84% for PLA+ parts. Additionally, regarding infill density, tensile strength ranged between 5.94 MPa at 15% infill and 13.28 MPa at 50% infill. Confidence in the obtained results was provided by the fact that the registered values and variation tendency were observed to fall in the range of values obtained by other authors with tests that were made using similar process parameters and manufacturing conditions on uncoloured parts [39,40,41,42]. The variation of mean effects highlighted that colour distribution on the filament deviated by up to 5.34% from 33% distribution to 99% distribution, and colour pigment deviated by 9.55% when using the red colour. Furthermore, in Figure 15, a comparison between coloured and uncoloured parts was made, and while the uncoloured parts occupy a median position, it can be seen that the results for coloured parts varied significantly, from which the main conclusion can be derived: the colouring of the filament has an impact on the tensile strength of 3D-printed parts.

In terms of surface quality, it was observed that by using the colouring method, the parts presented a better surface finish overall. A deviation of up to 6% was observed when the colour distribution was varied. Furthermore, colour pigment presented a deviation of up to 9% when the red colour was used. As seen in research specific to the additive manufacturing domain [43], better results were obtained when using PLA compared to PETG (up to 2.88%), and surface roughness improved with increasing infill density (up to 10.56% at 50% infill compared to 15% infill). This same aspect was highlighted by Figure 16, where it can be seen that PLA+ registered the smallest values for surface roughness, which was also observed in other studies [38]. Possibly due to a reaction between the amorphous structure of PETG and the alcohol-based ink, the overall values for surface quality were the worst to be presented, which can also be observed in the microscopic surface analysis where PETG presented more air pores overall, which is an effect possibly caused by improper evacuation of vapours, itself caused by filament moisture or ink alcohol.

The Charpy test values ranged from 0.3 J to 0.4 J and could highlight the nature of the impact. It was observed that both the coloured and uncoloured material had brittle fractures, an aspect specific to the studied materials. Surface inspection highlighted that, by using the colouring method, the obtained surfaces presented a series of impurities, such as air pores and wear debris. Overall, air pores were present more often when PETG was used. This effect may be due to the improper evacuation of vapours caused by filament moisture or ink alcohol. This observed factor could lead to lower tensile strength, a fact which was observed in testing experiments. 

In conclusion, although 3D in-process colour printing has a series of advantages and limitations, the results obtained are promising, and future tests will be carried out to investigate the implications of other input parameters (layer thickness, printing temperature, printing speed, etc.) on the capabilities of in-process colouring 3D printing. Furthermore, other materials and inks will be tested. 

## Figures and Tables

**Figure 1 polymers-14-05173-f001:**
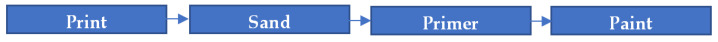
Conventional post-processing for the colourisation of 3D-printed parts.

**Figure 2 polymers-14-05173-f002:**
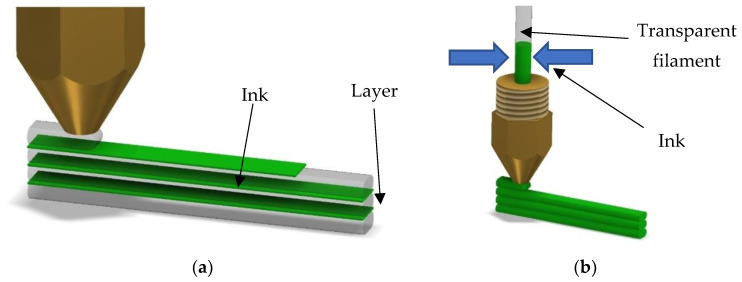
Main in-process possibilities for reproducing colours in 3D MEX processes: (**a**) layer colouring; (**b**) filament colouring.

**Figure 3 polymers-14-05173-f003:**
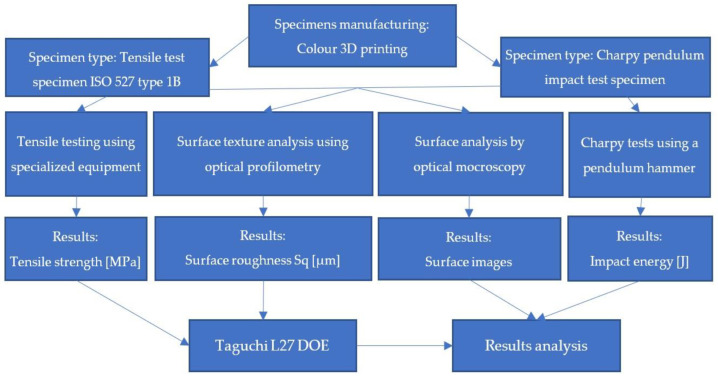
Experimental program.

**Figure 4 polymers-14-05173-f004:**
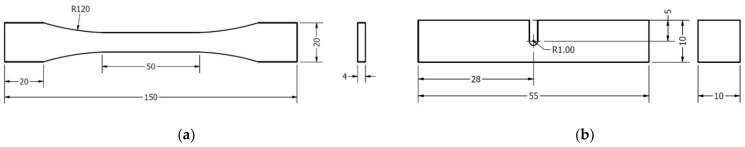
Testing tubes: (**a**) tensile testing tube ISO 527 type 1B; (**b**) Charpy testing tube.

**Figure 5 polymers-14-05173-f005:**
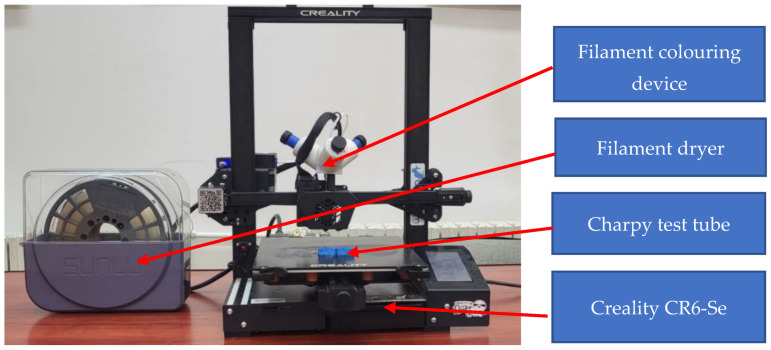
Fabrication setup.

**Figure 6 polymers-14-05173-f006:**
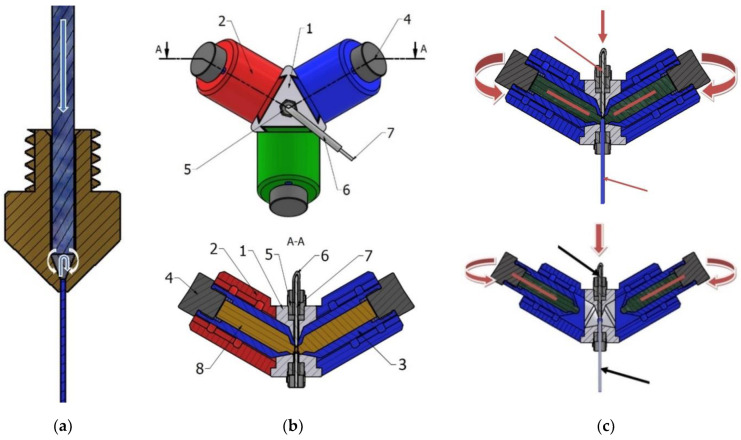
Colouring of the filament: (**a**) principle; (**b**) colouring device; (**c**) open and closed positions.

**Figure 7 polymers-14-05173-f007:**
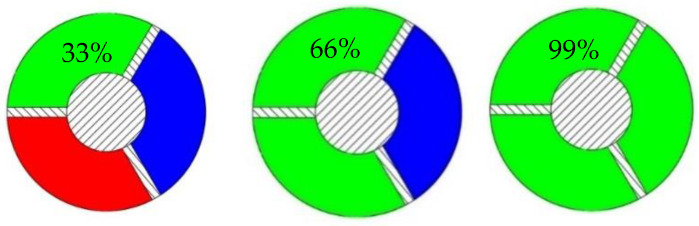
Percentage of filament colouring with the proposed device.

**Figure 8 polymers-14-05173-f008:**
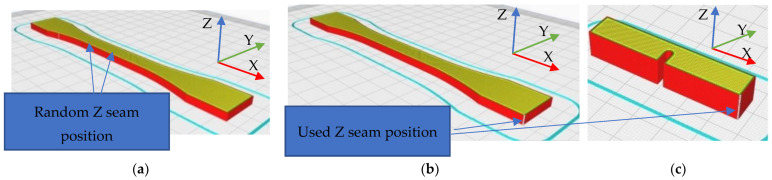
Z seam positions and printing direction: (**a**) random; (**b**,**c**) user-specified.

**Figure 9 polymers-14-05173-f009:**
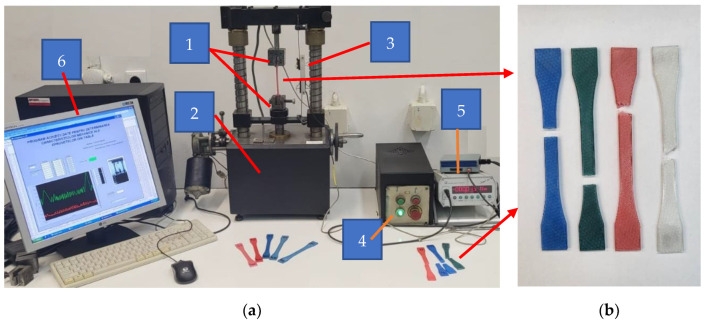
Tensile testing stand: (**a**) components; (**b**) tested specimens.

**Figure 10 polymers-14-05173-f010:**
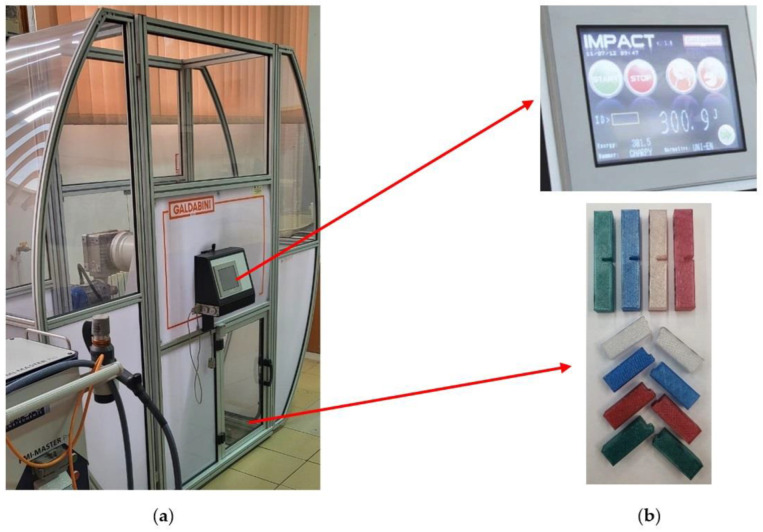
Tensile testing stand: (**a**) Galbadini Impact 300; (**b**) tested specimens and value recording.

**Figure 11 polymers-14-05173-f011:**
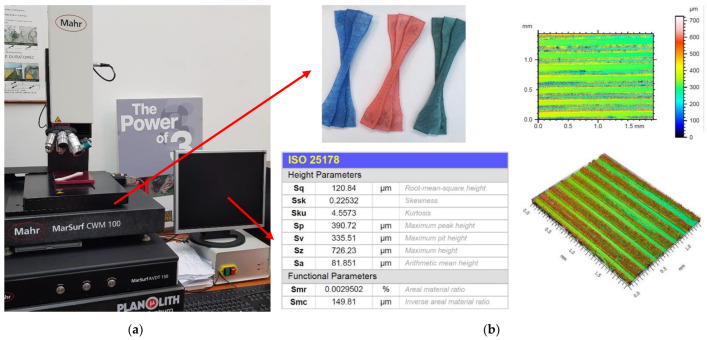
Surface roughness measurement: (**a**) Mahr MarSurf CWM 100; (**b**) tested specimens and value recording.

**Figure 12 polymers-14-05173-f012:**
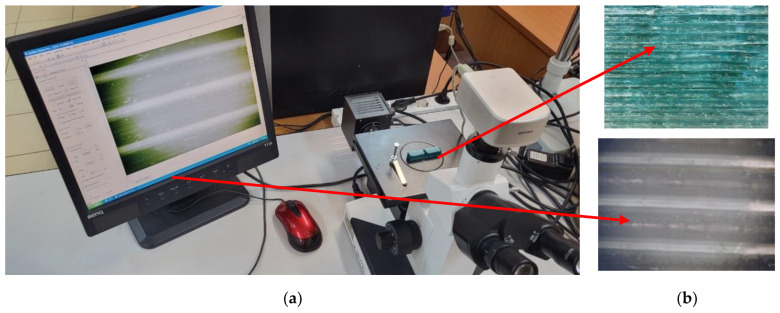
Surface analysis: (**a**) Optech microscope; (**b**) surface images.

**Figure 13 polymers-14-05173-f013:**
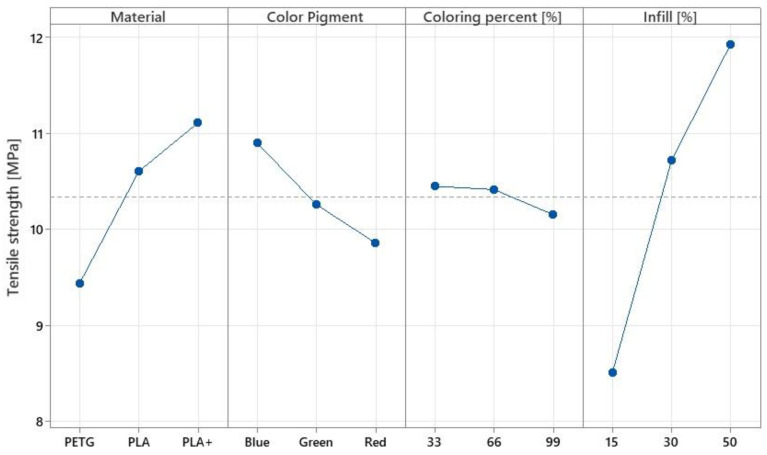
Variation of the main effects for tensile tests.

**Figure 14 polymers-14-05173-f014:**
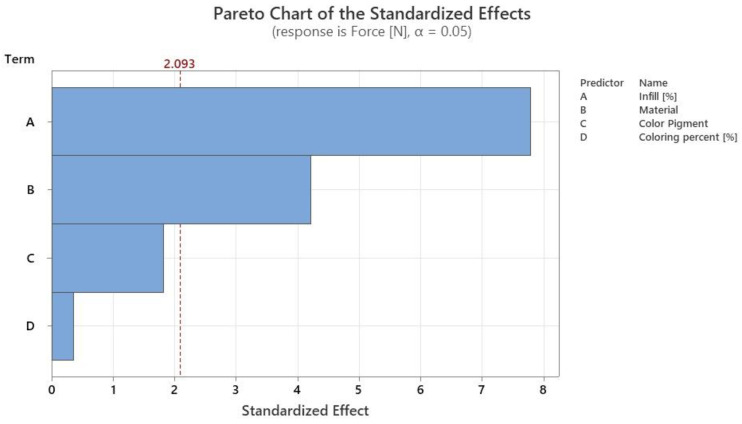
Pareto graph for the significance level of the studied factors for tensile tests.

**Figure 15 polymers-14-05173-f015:**
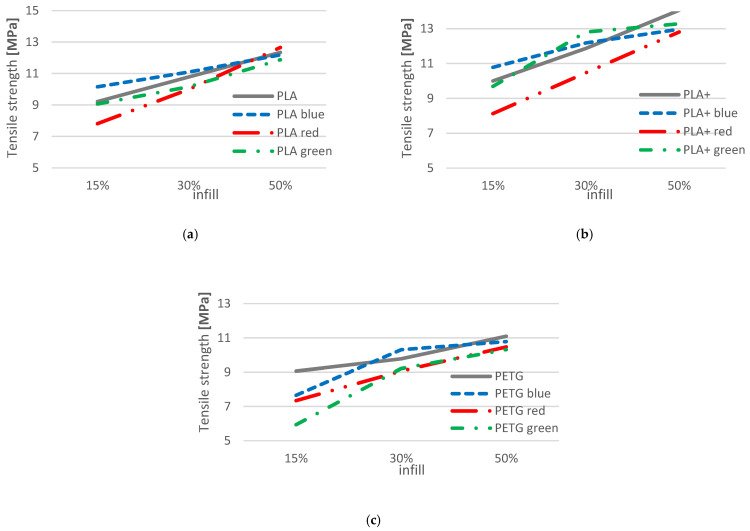
Comparison of tensile strength between coloured and uncoloured printed parts: (**a**) PLA material; (**b**) PLA+ material; (**c**) PETG material.

**Figure 16 polymers-14-05173-f016:**
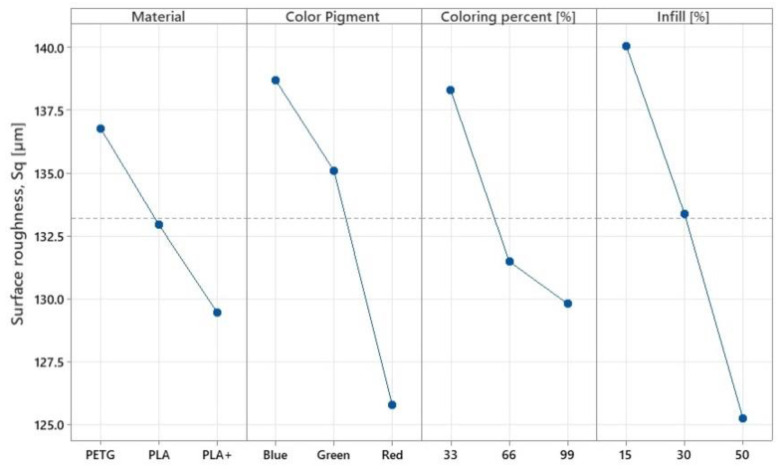
Variation of main effects for surface roughness.

**Figure 17 polymers-14-05173-f017:**
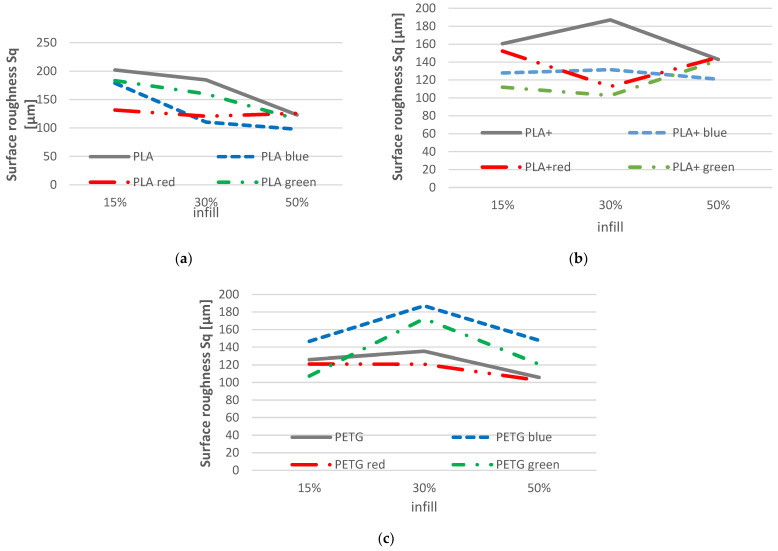
Comparison of surface roughness between coloured and uncoloured printed parts: (**a**) PLA material; (**b**) PLA+ material; (**c**) PETG material.

**Figure 18 polymers-14-05173-f018:**
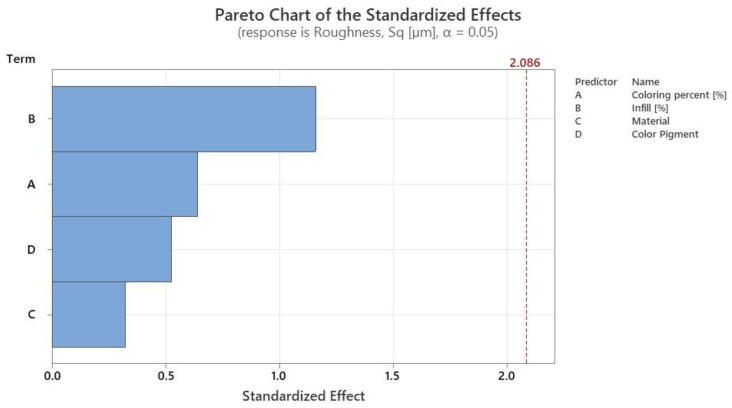
Pareto graph for the significance level of the studied factors for surface roughness.

**Figure 19 polymers-14-05173-f019:**
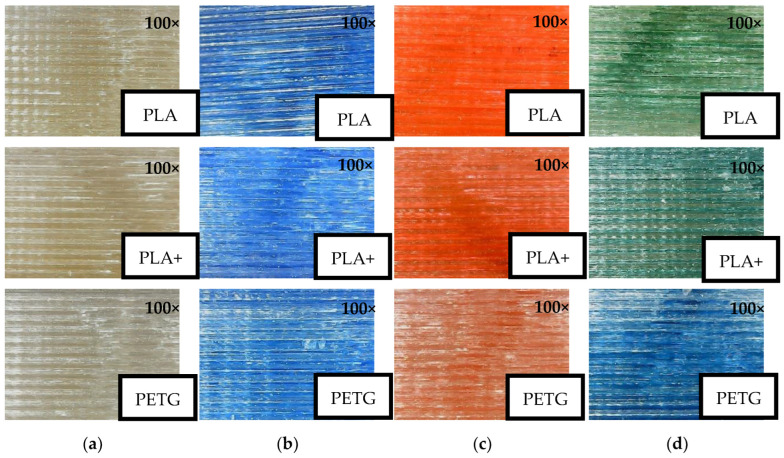
Surface analysis: (**a**) uncoloured filament; (**b**) blue pigment; (**c**) red pigment; (**d**) green pigment.

**Figure 20 polymers-14-05173-f020:**
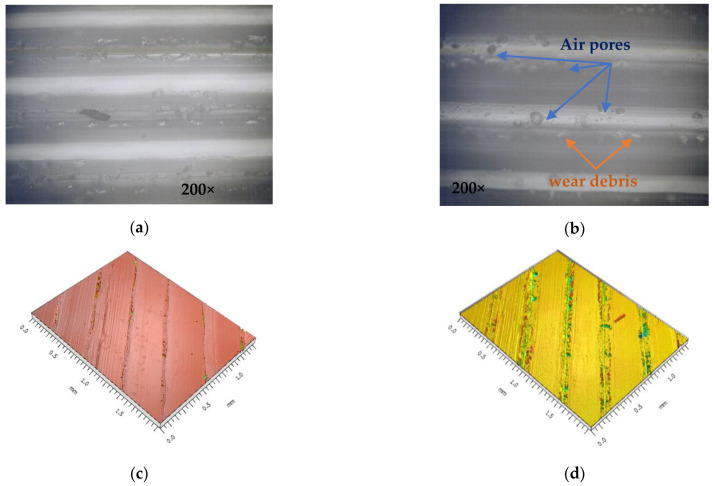
Surface analysis: (**a**,**c**) PLA, blue ink, colour distribution 33%, infill 15%; (**b**,**d**) PLA+, green ink, colour distribution 33%, infill 15%.

**Figure 21 polymers-14-05173-f021:**
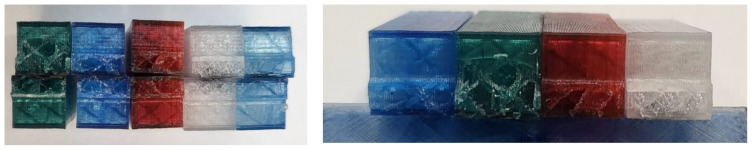
Charpy test specimens.

**Table 1 polymers-14-05173-t001:** Taguchi L27 DOE.

Test Number	Material	Colour Pigment	Colour Distribution [%]	Infill [%]	Tensile Strength, [MPa]	Roughness, Sq [µm]
1	PLA	Blue	33	15	10.16	178.76
2			66	30	11.09	110.33
3			99	50	12.19	97.65
4		Red	33	30	10.00	120.65
5			66	50	12.66	125.77
6			99	15	7.810	131.52
7		Green	33	50	11.88	115.55
8			66	15	9.060	183.24
9			99	30	10.16	160.05
10	PLA+	Blue	33	30	12.19	131.56
11			66	50	12.97	120.84
12			99	15	10.78	127.75
13		Red	33	50	12.81	145.23
14			66	15	8.120	152.28
15			99	30	10.47	112.74
16		Green	33	15	9.690	111.91
17			66	30	12.81	102.70
18			99	50	13.28	141.84
19	PETG	Blue	33	50	10.78	147.61
20			66	15	7.660	146.64
21			99	30	10.31	187.05
22		Red	33	15	7.340	120.98
23			66	30	9.060	120.68
24			99	50	10.47	102.24
25		Green	33	30	9.220	172.40
26			66	50	10.31	120.84
27			99	15	5.940	107.31

**Table 2 polymers-14-05173-t002:** Uncoloured test values.

Material	Infill [%]	Tensile Strength, [MPa]	Roughness, Sq [µm]
PLA	15	9.220	202.00
PLA	30	10.78	184.44
PLA	50	12.34	123.01
PLA+	15	10.00	160.53
PLA+	30	11.88	187.07
PLA+	50	14.06	143.00
PETG	15	9.060	126.00
PETG	30	9.780	135.35
PETG	50	11.09	105.76

**Table 3 polymers-14-05173-t003:** Material proprieties.

Material	Density [g/cm^3^]	Heat Distortion Temp [°C, 0.45 MPa]	Melt Flow Index [g/10 min]	Flexural Modulus [MPa]
PLA	1.2	53	3.5	1915
PLA+	1.23	53	5	1973
PETG	1.27	64	20	1073

**Table 4 polymers-14-05173-t004:** Process parameters.

Settings	PLA/PLA+	PETG
Temperature	200 °C	240 °C
Bed temperature	60 °C	80 °C
Layer height	0.2 mm	
Fan	100%	
Retraction	6.5 mm	
Speed	50 mm/s	
Infill type	Cubic	
Wall line count	4	
Nozzle diameter	Φ0.4 mm	

**Table 5 polymers-14-05173-t005:** Tensile strength variation degree.

Values	Material	Colour Pigment	Colour Distribution	Infill
1	-	-	-	-
2	↑ 4.77%	↓ 5.89%	↓ 0.33%	↑ 26.00%
3	↓ 11.01%	↓ 9.55%	↓ 5.34%	↑ 40.28%

**Table 6 polymers-14-05173-t006:** Uncoloured test values/tensile strength.

Material	Infill [%]	No Colour	1	2	3	Maximum Variation
PLA	15	9.220	10.16	7.810	9.060	↓ 15.24%
PLA	30	10.78	11.09	10.00	10.16	↓ 7.220%
PLA	50	12.34	12.19	12.66	11.88	↓ 3.790%
PLA+	15	10.00	10.78	8.120	9.690	↓ 18.75%
PLA+	30	11.88	12.19	10.47	12.81	↓ 11.84%
PLA+	50	14.06	12.97	12.81	13.28	↓ 8.880%
PETG	15	9.060	7.660	7.340	5.940	↓ 34.48%
PETG	30	9.220	10.16	7.810	9.060	↓ 7.350%
PETG	50	10.78	11.09	10.00	10.16	↓ 7.040%

**Table 7 polymers-14-05173-t007:** Degree of variation for Sq roughness.

Values	Material	Colour Pigment	Colour Distribution	Infill
1	-	-	-	-
2	↓ 2.62%	↓ 2.59%	↓ 4.93%	↓ 4.89%
3	↑ 2.88%	↓ 9.3%	↓ 6.15%	↓10.56%

**Table 8 polymers-14-05173-t008:** Uncoloured test values/Sq roughness.

Material	Infill [%]	No Colour	1	2	3	Maximum Variation
PLA	15	202.00	178.76	131.52	183.24	↓ 34.89%
PLA	30	184.44	110.33	120.65	160.05	↓ 40.18%
PLA	50	123.01	97.65	125.77	115.55	↓ 20.62%
PLA+	15	160.53	127.75	152.28	111.91	↓ 30.29%
PLA+	30	187.07	131.56	112.74	102.70	↓ 39.73%
PLA+	50	143.00	120.84	145.23	141.84	↓ 15.50%
PETG	15	126.00	146.64	120.98	107.31	↑ 16.38%
PETG	30	135.35	187.05	120.68	172.4	↑ 38.20%
PETG	50	105.76	147.61	102.24	120.84	↑ 39.57%

## Data Availability

Some or all data, models, or code generated or used during the study are available from the corresponding author by request.

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
