# Peer review of "Experimental Study on the Possibilities of FDM Direct Colour Printing and Its Implications on Mechanical Properties and Surface Quality of the Resulting Parts"

_polymers, 2022, doi:10.3390/polym14235173_

Round 1

Reviewer 1 Report

1.            It is necessary to explain how and why the Tensile force was measured. In accordance with the ISO 527-1 standard, the tensile properties of plastics are defined: tensile strength, modulus of elasticity, etc. Give literary references to the ISO 527 standards used.

2.            It is recommended to replace all values of Tense force with the corresponding values of tense strength.

3.            It is recommended to provide data on the modulus of elasticity.

4.            In line 116, specify the name of the manufacturer's company.

5.            In line 117, give a reference to technical data sheets.

6.            It is recommended to describe the printing direction in lines 173-175.

7.            Check the designation of Figure 6 (c) in Figure 6.

8.            Specify the manufacturer of the Tense testing stand.

9.            Indicate in Figure 13 which property (strength, modulus of elasticity) is shown and the units of measurement.

10.          It is recommended to discuss how the results obtained can be applied in the engineering practice of manufacturing products by 3D printing.

Author Response

Response to Reviewer 1 Comments

Point 1. It is necessary to explain how and why the Tensile force was measured. In accordance with the ISO 527-1 standard, the tensile properties of plastics are defined: tensile strength, modulus of elasticity, etc. Give literary references to the ISO 527 standards used.

Response 1: Tensile force data was replaced by tensile strength data, determined in accordance to the ISO 527-1 standard. References to the ISO 527 standards used was added ([29, 30]). See revised paper.

Point 2. It is recommended to replace all values of Tense force with the corresponding values of tense strength.

Response 2: Tensile force data was replaced by tensile strength data, determined in accordance to the ISO 527-1 standard. See revised paper.

Point 3. It is recommended to provide data on the modulus of elasticity.

Response 3: The technical datasheets provided by the material manufacturer do not offer information on Young’s modulus of elasicity. Only the Flexural modulus was given, and information regarding that parameter was added in Table 3. References to the materials datasheets were also added. See revised paper.

Point 4. In line 116, specify the name of the manufacturer's company.

Response 4: The investigated filaments were manufactured by eSUN - Shenzhen Esun Industrial Co., Ltd., details and references were added in revised paper.

Point 5. In line 117, give a reference to technical data sheets.

Response 5: Refences to the technical data sheets are given in the revised paper.

Point 6. It is recommended to describe the printing direction in lines 173-175.

Response 6: Printing direction was described in the revised paper. Also, a coordinate system was added to figure 8.

Point 7. Check the designation of Figure 6 (c) in Figure 6.

Response 7: figure designation was changed

Point 8. Specify the manufacturer of the Tense testing stand.

Response 8: The tensile strength of the specimens (Figure 9.b) was measured using an experimental set-up (Figure 9.a), previously built within the Faculty of Mechanical Engineering, Automotive and Robotics from the “Stefan cel Mare” University of Suceava, Romania.

Point 9. Indicate in Figure 13 which property (strength, modulus of elasticity) is shown and the units of measurement.

Response 9: Figure 13 was modified, see revised paper.

Point 10. It is recommended to discuss how the results obtained can be applied in the engineering practice of manufacturing products by 3D printing.

Response 10: Conclusions were modified, see revised paper for details.

Reviewer 2 Report

The article "Experimental study on the possibilities of FDM direct colour printing and its implications on mechanical properties and surface quality of the resulted parts" presents experimental tensile and surface quality of  PAL, PLA+, PETG at three diff factors and three different levels. The presentation of the article is good and the authors had conducted reasonable number of tests. However the article can be further improved by consider the following comments. 

1) Please confirm whether ref 22 is used in this article.

2) line 87- "3 different pigments"- since the main focus of this paper is the colour, it will be great if more details of pigments can be provided, in order for others to repeat your results.

3) Table 3, please confirm the melt flow index for PLA, is 3.5 (or 3, 5?)

4) Fig 4, please provide units.

5) Please provide the printing orientation for impact sample.

6) Fig 6, ink reservoir (3) is not shown in your Fig.

7) line 198, please provide cutoff length for surface roughness measurement, and location/direction of measurement.

8) Line 252, please cite the source for 'confirming other studies in the literature (???).

9) line 259, please further explain "convective heat transfer coefficient decreases ...."

Author Response

Response to Reviewer 2 Comments

Dear reviewer,

Thank you for taking the time to consider our paper, and for the offered suggestions.

Point 1. Please confirm whether ref 22 is used in this article.

Response 1: Reference 22 was indeed ommited in the first paper manuscript. The error was remedied, see revised paper.

Point 2. line 87- "3 different pigments"- since the main focus of this paper is the colour, it will be great if more details of pigments can be provided, in order for others to repeat your results.

Response 2: The ink used for the printing process was a highly saturated, fast-drying, highly transparent alcohol-based ink, produced by Jacquard, USA, in the Piñata Alcohol Ink line of products (see https://www.jacquardproducts.com/pinata-alcohol-ink). The manufacturer does not offer detailed data on the exact chemical composition, as it is a trade secret.

Point 3. Table 3, please confirm the melt flow index for PLA, is 3.5 (or 3, 5?)

Response 3: The value of the melt flow index was corrected in table 3 (it should read 3.5 g/10min).

Point 4. Fig 4, please provide units.

Response 4: In Figure 4, the specified dimensions are in mm (the metric system was used). This is now mentioned in the text above figure 4.

Point 5. Please provide the printing orientation for impact sample.

Response 5: Printing orientation was described in the revised paper. Also, a coordinate system was added to figure 8.

Point 6. Fig 6, ink reservoir (3) is not shown in your Fig.

Response 6: Figure 6 b was modified, it now shows the ink reservoir (3).

Point 7. line 198, please provide cutoff length for surface roughness measurement, and location/direction of measurement.

Response 7: Measurements were taken in the central region of the specimens, on 2x1.5 mm areas. Surface roughness was considered, so both directions are taken into account. The parameters were detemined in accordnce to ISO25178 Standard, and with the default settings of the MountainsLab 8.1 software, which uses as predifined, a 0.8 mm Robust Gaussian L-filter. The above information was added in the revised paper.

Point 8. Line 252, please cite the source for 'confirming other studies in the literature.

Response 8: Reference [38] was added, see revised paper.

Point 9. line 259, please further explain " convective heat transfer coefficient decreases ...."

Response 9: Additional explanations were given in the revised paper. See lines 276-281 in the revised paper.

Reviewer 3 Report

The manuscript is very interesting and the research is designed properly. Unfortunately, it has a lot of issues which I mentioned below:

1. Please use proper terminology of the additive manufacturing technology (material extrusion) which is sanctioned by the ISO/ASTM 52900 standard.

2. You put too much general information in the abstract part. You have to rewrite it and include the most important outcomes of your work. 

3. Please highlight what is the novelty of your research in comparison to the present state of the art (what you have in your research that is new and original).

4. Replace tensile force (N) with tensile strength (MPa) everywhere. 

5. What is PLA+? How does it differ from pure PLA? You have to put the material description first (lines 122-125 have to be moved near lines 87-89)

6. "...produced from starch rich crops such as corn, sugarcane, potato or maize." Such kind of material description is unacceptable. Put proper, scientific descriptions with the chemical composition of each material. 

7. Line 116 - put the source of the material (producer data).

8. Figure 3 - what is "Mahr CWM 100? or "Optika" Microscope? Please clarify this graph. 

9. Please include the name of the producer of the coloring device. 

10. Lines 142  - 145 - it is hard to understand this part. Please rewrite it to make it clear and easy to understand. 

11. How many test samples have you used for each test and how does it correspond with the used standard? 

12. What is the unit in Mean of Means? Please better describe those two charts (F.13 and F.16)

13. Table 8 - fix the mess with mixed dots and commas. 

14. There is not any quantified value in the conclusion part. Please rewrite all conclusions and make them point-by-point. 

After correction of the mentioned issues and significant improvements, the paper could be reconsidered for acceptation. 

Author Response

Response to Reviewer 3 Comments

Dear reviewer,

Thank you for taking the time to consider our paper, and for the offered suggestions.

Point 1. Please use proper terminology of the additive manufacturing technology (material extrusion) which is sanctioned by the ISO/ASTM 52900 standard.

Response 1: The terminology was modified throughout the paper in accordance to the ISO/ASTM 52900 standard. See revised paper.

Point 2. You put too much general information in the abstract part. You have to rewrite it and include the most important outcomes of your work.

Response 2: The abstract was rewritten, hopefully it is more adequate in the revised version of the paper.

Point 3. Please highlight what is the novelty of your research in comparison to the present state of the art (what you have in your research that is new and original).

Response 3: The novelty of the present research was highlighted in the revised paper. See revised paper, lines 77-83.

Point 4. Replace tensile force (N) with tensile strength (MPa) everywhere.

Response 4: Tensile force data was replaced by tensile strength data, throughout the paper, determined in accordance to the ISO 527-1 standard. See revised paper.

Point 5. What is PLA+? How does it differ from pure PLA? You have to put the material description first (lines 122-125 have to be moved near lines 87-89)

Response 5: A brief description of PLA+ was added in the introduction section, (see lines 87,88) of the revised paper. Additional information on all the used materials, including additional technical data were detailed in section “2.1Materials”. Supplementary references were also added.

Point 6. "...produced from starch rich crops such as corn, sugarcane, potato or maize." Such kind of material description is unacceptable. Put proper, scientific descriptions with the chemical composition of each material.

Response 6: Additional information on all the used materials, including chemical composition data, as offered by the manufacturer, were added in section “2.1Materials”. See revised paper lines 117-125

Point 7. Line 116 - put the source of the material (producer data).

Response 7: Aditional informations were given in section “2.1Materials”. References to the manufacturer’s technical data were added (see references [31-36]).

Point 8. Figure 3 - what is "Mahr CWM 100? or "Optika" Microscope? Please clarify this graph.

Response 8: Figure 3 was modified and additional explanations were given (see lines 102-108) in the revised paper.

Point 9. Please include the name of the producer of the coloring device. 

Response 9: The coloring device is of own design and construction, by the authors. This information was added in the revised paper.

Point 10. Lines 142  - 145 - it is hard to understand this part. Please rewrite it to make it clear and easy to understand.

Response 10: Lines 142 - 145 were reformulated. See revised paper, lines 150--154.

Point 11. How many test samples have you used for each test and how does it correspond with the used standard? 

Response 11: The tests were conducted in accordance with ISO 93 527 standard: Plastics - Determination of tensile properties, part 1 and 2. References [29, 30] were added in the revised paper. In accordance to the mentioned standard, the minimum number of 5 test samples were used for each test, and the mean values were further considered when the tables and graphs were created.

Point 12. What is the unit in Mean of Means? Please better describe those two charts (F.13 and F.16)

Response 12: Figures 13 and 16 were modified in the revised paper.

Point 13. Table 8 - fix the mess with mixed dots and commas.

Response 13: Tables were modified, so that decimals are consistently represented by dots throughout the paper. Se revise paper

Point 14. There is not any quantified value in the conclusion part. Please rewrite all conclusions and make them point-by-point.

Response 14: The conclusions section was rewritten. See revised paper.

Round 2

Reviewer 1 Report

The manuscript can be published in this form.

Author Response

Dear reviewer,

Thank you for taking the time to consider our paper, and for the offered suggestions.

Reviewer 2 Report

The authors have addressed all previous suggestions/comments.

Thank you.

Author Response

Thank you for taking the time to consider our paper, and for the offered suggestions.

Reviewer 3 Report

The authors made almost all the improvements. - there are still some issues with figure 3 which I mentioned in my 1st review. Please make it clear and after that, in my opinion, the paper could be accepted for publication. 

Author Response

Response to Reviewer 3 Comments

Dear reviewer,

Thank you again for taking the time to consider our paper, and for the offered suggestions.

 Point 1. The authors made almost all the improvements. - there are still some issues with figure 3 which I mentioned in my 1st review. Please make it clear and after that, in my opinion, the paper could be accepted for publication.

Response 1: Figure 3 was further modified and the following description was added in the revision2 version of the paper:  “Figure 3 graphically illustrates the steps taken for the present experimental study, which are further described. Two types of specimens were printed, one for tensile tests, and one for impact strength testing. Tensile tests were conducted using specialized equipment, and Charpy tests were performed using a Galdabini Impact 300 pendulum hammer. Surface micro-topographies of test specimens were mapped using a Mahr MarSurf CWM 100 confocal microscope and interferometer, and surface quality parameters were determined. Optical images of the surfaces were taken by aid of an "Optech" model IM/IMT microscope and the Optika Vision Pro image analysis software. All the abovementioned experimental steps and equipment are further described in section 2.2.”.

 See revision 2 of the paper.